Comparative study of the antibacterial effects of wound secretions of different cultivars of Chinese fir

Jiang Yu
Zeng Yalin
Zhu Jianing
Sun Linjun
Wu Pengfei
Li Ming
Ma Xiangqing lxymxq@126.com
1 College of Forestry, Fujian Agriculture and Forestry University , Fuzhou , Fujian , China
2 National Forestry and Grass Bureau of Chinese Fir Engineering Technology Research Center , Fuzhou , Fujian , China
Kafle Dr. Gandhiv
Electronic publication date: 2024 Aug 16
Publication date: 2024
Volume: 12
Electronic Location ID: e17850
Received 2024 Mar 18; Accepted 2024 Jul 11
Copyright: ©2024 Jiang et al.
Copyright year: 2024
Copyright holder: Jiang et al.
License: This is an open access article distributed under the terms of the Creative Commons Attribution License, which permits unrestricted use, distribution, reproduction and adaptation in any medium and for any purpose provided that it is properly attributed. For attribution, the original author(s), title, publication source (PeerJ) and either DOI or URL of the article must be cited.
License URL: https://creativecommons.org/licenses/by/4.0/

Keywords: Chinese fir, Antibacterial effect, Wound secretions, Cultivars, Bacterial minimum inhibitory concentration

Funding: National Key Research and Development Program of China 2021YFD2201302 This work was supported by the National Key Research and Development Program of China (2021YFD2201302). The funders had no role in study design, data collection and analysis, decision to publish, or preparation of the manuscript.

==============================
Background

The bark of Chinese fir (Cunninghamia lanceolata), the largest afforestation tree species in the forest areas of southern China, is susceptible to injuries and bites from small animals. The population of small animals has recently increased owing to improvements in the ecological environment across various forested areas, thus increasing the incidence of injuries in the bark of Chinese fir. Following such injuries, the bark secretes light yellow or milky white secretions, the function of which remains unclear. The present study aimed to reveal the antibacterial effect of exudates of different Chinese fir cultivars on five bacterial species.

Methods

The research involved three-year-old plantations of Taxus chinensis var. koraiensis and Yangkou3 and three-year-old container plantations of Taxus chinensis var. pendula, Yang 061, and Yang 020. The antibacterial effects of exudates were analyzed using the filter paper diffusion method. The minimum inhibitory concentration for each secretion and the bacterial inhibition zone were determined.

Results

The exudates of the different Chinese fir bark exhibited notable antibacterial effects on Bacillus subtilis, Salmonella paratyphi B, Pseudomonas aeruginosa, Escherichia coli, and Staphylococcus aureus. However, the extent of these antibacterial effects varied among the different Chinese fir cultivars, as the minimum inhibitory concentrations (MICs) of the exudates against the five bacterial species varied. The mean MIC of Pseudomonas aeruginosa was lower potency, whereas that of Escherichia coli was the lowest. Notably, the antibacterial efficacy of the exudates was mainly influenced by the composition of the secretions rather than the number of secretions, with organic acid compounds and terpenoids potentially contributing to the antibacterial effects against E. coli and Bacillus subtilis, respectively.

Conclusion

This study demonstrates the antibacterial effect of wound secretion of different Chinese fir cultivars, highlighting their varying efficacy on different bacterial species. Moreover, the antibacterial ability of the exudates of the strains was mainly determined by the composition of the wound secretions, and there was no noticeable relationship with the number of wound secretions. The results of this study offers a theoretical basis for screen Chinese fir cultivars with high-disease-resistant.

Introduction

Following injury, many plants secrete various phenolic secondary compounds and terpenoids of diverse colors. These secretions have different ecological functions (Scalarone, Lazzari & Chiantore, 2003; Echard & Lavédrine, 2008). They rapidly cover the wound and play a crucial role in sealing and protecting the wound, thus preventing the invasion of foreign organisms (Zaheri-Abdevand & Badr, 2023). Numerous studies have shown that shikimic acid, isovitexin, isopimaric acid, and epicatechin, which can be extracted from plant wound secretions (Hu et al., 2021; Azubuike-Osu et al., 2021; Liu et al., 2022; Cao, 2023; Hid et al., 2023; Panagiotidou et al., 2023), exhibit various pharmacological effects, including antioxidant, anti-inflammatory, and antibacterial effects. Therefore, research on plant wound secretion holds substantial practical significance.

Chinese fir (Cunninghamia lanceolata) is the main afforestation tree species in the forest areas of southern China (Ma et al., 2003), renowned for its rapid growth, high yield, and good material quality. In recent years, the population of small animals has increased owing to improvements in the ecological environment across various forested areas. Unfortunately, in some areas, the bark of Chinese fir forests is vulnerable to small animal bites and injuries, and following such incidents, the bark secretes light yellow or milky white secretions. However, the function of these wound secretions on the Chinese fir remains unclear. These wound secretions are commonly referred to as exudates. We have previously demonstrated that different varieties of Chinese fir exhibit different environmental response characteristics developed over long-term adaptation to different environments. Notably, there are significant differences in the number and composition of exudates following dry skin injury among different varieties of Chinese fir (Zeng et al., 2023). However, there is a lack of understanding of the biological functions of the secretion of different varieties and cultivars of Chinese fir wound secretions, particularly regarding whether they possess antibacterial properties. This knowledge gap limits the planning and management of artificial afforestation of different Chinese fir varieties.

Bacillus subtilis is a plant growth-promoting rhizobacteria. Its physiological characteristics are rich and diverse, and it plays an important role in the prevention and control of crop diseases (Li et al., 2023). Pseudomonas aeruginosa is a common bacteria after wound infection in nature (Wang et al., 2020). Salmonella paratyphi, Escherichia coli, and Staphylococcus aureus are major foodborne pathogens (Lu et al., 2016). Several recent studies have shown that plant extracts and chemical substances in some foods have antibacterial effects on the five strains mentioned above (Fu et al., 2021; Matcheme et al., 2023; Srivastava et al., 2023). However, the antibacterial effect of Chinese fir exudates is less involved, and how these exudates make Chinese fir disease resistance is still unclear. In the 1960s, the American Institute of Water Oncology performed antitumor bioassays on plant extracts (Zhou, 1978). Since then, the extraction of natural active antibacterial substances from plants has attracted significant attention (Fang & Jiang, 2022; Mekonnen Bayisa & Seid Bultum, 2022; Yu, Li & Tang, 2023). Plant extracts, such as honeysuckle, exhibit inhibitory effects on various bacterial species, including Bacillus subtilis, Pseudomonas aeruginosa, Escherichia coli, Staphylococcus aureus (Huang et al., 2016; Zhou & Tang, 2018; Jeong et al., 2023). In addition to their antibacterial role, these extracts often contain terpenes, alkaloids, flavonoids, phenols, and glycosides (Tewari & Gomatinayagam, 2023). However, the antimicrobial activity of exudates of Chinese fir has not been well studied.

In the present study, we utilized different cultivars of Chinese fir plantations established in 2019 as the research focus. We collected exudates through an artificially simulated bark mechanical damage test and investigated the antibacterial effect of these exudates on five different bacterial species using the filter paper diffusion method. We compared differences in the antibacterial effects of the exudates and determined their minimum inhibitory concentration (MIC) to provide a scientific basis for screening high-disease-resistant Chinese fir varieties. We hypothesized that (1) the exudates of five varieties of Chinese fir would exhibit antibacterial effects on Bacillus subtilis, Salmonella paratyphi B, Pseudomonas aeruginosa, Escherichia coli, and Staphylococcus aureus, and (2) varieties with similar compositions of exudates would exhibit similar antibacterial capacity.

Materials & Methods

Overview of the test site

The experimental site is situated in the Yangkou State-owned Forest Farm in Shunchang County, Fujian Province (26°50′N, 117°53′E). The location is categorized as grade II, featuring mountain red soil and an elevation of 260 m. With a mid-subtropical maritime monsoon climate, it has an average annual temperature of 18.5 °C, an average annual sunlight duration of 1740 h, an average annual frost-free period of 230 d, and an average annual precipitation of 1,880 mm. In 2019, various experimental Chinese fir forest strains were cultivated. Chinese fir clonal container plantation Yang 061 and Yang 020 were grown in the Forest Farm of Fujian Province for afforestation purposes. Hubei Chuizhisha developed a clonal container plantation that matured into Chinese firs. Yangkou 3 is a cultivated at the Forest Farm of Fujian Province, owned by Yangkou State, using seeds from a Chinese fir seed orchard dating back to the third generation. Red-heart Chinese fir were cultivated at the Jiangxi Province’s Chenshan Forest Farm in Anfu County.

Experimental design

Artificial afforestation of different Chinese fir cultivars was performed in February 2019 using a randomized block design. Each experimental plot, ranging from the foot of the mountain to the top of the mountain strip afforestation, covered an area of more than 500 m2, with Schima Superba as the isolation zone. The planting density for each experimental plot was 3,000 plants hm−2. A total of three blocks and 15 experimental plots were constructed. The growth of different cultivars in 3a Chinese fir plantations is shown in Table 1.

Table 1 Growth of Chinese fir plantations of different cultivars.

Chinese fir cultivars	DBH/cm	Tree height/m	
Chuizhisha	3.35 ± 0.11	3.22 ± 0.07	
Yangkou 3	4.78 ± 0.06	3.49 ± 0.11	
Red-heart	4.22 ± 0.01	3.65 ± 0.07	
Yang 020	3.47 ± 0.62	3.98 ± 0.35	
Yang 061	4.27 ± 0.53	3.76 ± 0.26	

Fifteen typical standard trees were selected in October 2021 for artificial mechanical damage wound simulation treatment in each cultivar of Chinese fir in a three-year-old experimental forest. Sharp knives were used to cut ellipses with a long half-axis of three cm and a short half-axis of one cm in the four directions of the bark in the east, west, south, north and south. The angle between the long half-axis of the ellipse and the horizontal line was 45∘, thus simulating the mechanical damage of the Chinese fir bark. Each strain of Chinese fir simulated mechanical damage treatment of 15 trees. After the bark was cut, 2 ml centrifuge tubes were placed at the lower part of the bark ellipse to collect wound secretions, and 4 centrifuge tubes were collected for each average tree of Chinese fir (Fig. 1). Each strain of Chinese fir collected 15 average wood exudates for 1 h, and the collection time was between 8:00 and 10:00 every morning. The samples were placed in dry ice at 4 °C and returned to the laboratory for further analysis.

Figure 1 Chinese fir wound secretion collection schematic diagram.

Photo credit: Zeng Yalin.

Based on the management status (Li et al., 2021; Tian et al., 2022) regarding pests and diseases of Chinese fir plantations, Bacillus subtilis, Salmonella paratyphi B, Pseudomonas aeruginosa, Escherichia coli, and Staphylococcus aureus were selected as the test species. Test bacteria were purchased from Beijing Lvyuan Bird Biotechnology Co., Ltd. The antibacterial effects of the collected exudates of Chinese firs were to test against different bacterial species.

Test treatment implementation

The antibacterial effects of exudates were ascertained using the filter paper diffusion method. A total of 3 g of beef extract, 5 g of sodium chloride, and 10 g of agar powder (20 g) were weighed using an electronic balance and placed in a 1,000-mL beaker. Ultrapure water was then added to the beaker, and the mixture was heated to slight boiling until the solids completely dissolved. The pH was then adjusted to 7.0, the mixture was sterilized for 30 min at 121 °C, and the mixture was refrigerated at 4 °C.

The preparation procedure for the liquid medium without the agar powder was the same as that for the solid medium. Test tubes were subpackaged after processing. The bacterial which grow in the sterile water stock solution was obtained by removing the dry powder test bacteria from the refrigerator at 4 °C, adding them to a test tube filled with sterile water, and shaking the mixture in a constant temperature (37 °C) shaker on the sterile ultra-clean bench. An appropriate amount of bacterial stock solution was inoculated onto the plate and activated in a constant temperature incubator at 37 °C for 24 h. The activated bacterial plate was inoculated with an inoculation rod, and the strain was transferred to a test tube containing sterile water. The bacterial concentration was diluted to 105 to 108 CFU/mL using a McBurney turbidity tube and stored in a refrigerator at 4 °C for later use. According to the above conditions, select the appropriate concentration of bacterial solution, if the bacteria can grow throughout the whole dish and the distribution is uniform. The sensitivity determination of the test antibacterial ring is shown in Table 2 (Wang & Zhang, 2023).

Table 2 Sensitivity determination of bacteriostatic ring test.

	Inhibition zone diameter (mm)	
	0	<10	10–14	15–20	>20	
Sensitivity	Insensitivity	Slightly	Moderately	Highly	Extremely	

Determination of antibacterial effect

One hundred microliters of the bacterial suspension were placed on a plate and evenly coated with a coating rod. sterile water soaked by exudates (6 mm in diameter) was attached to a solid plate, and sterile water infiltration filter paper was used as the blank control (CK). The prepared plates were placed in a constant temperature incubator at 37 °C for 24 h. The plate was removed, and the diameter of the inhibition zone was measured using a Vernier caliper. Each experiment was performed in triplicates.

Determination of MIC

The MIC was determined following a previously established method by Ning et al. (2022). Briefly, sterile water was used as the diluent, and six concentration gradients of wound secretion were prepared at 8, 4, 2, 1, 0.5, and 0.25 mg/mL. A total of 100 µL of bacterial suspension was added to 9 mL of a liquid medium containing exudates and thoroughly mixed. After incubation in a constant temperature incubator for 24 h, the plate was inoculated and placed in a constant temperature incubator for an additional 24 h to observe colony growth on each plate. Each concentration gradient was repeated three times, with sterile water as the control, and the presence of growth inhibition was used as to determine the antibacterial activities of the wound secretions.

Data processing and analysis

SPSS software (version 24.0) was used for statistical analysis. Single factor T-test was used in RStudio to compare the effects of exudates of five strains of Chinese fir on different strains (α = 0.05). Excel 2010 software was used to create the charts.

Results

Antibacterial effects of wound secretions from different Chinese fir cultivars on Bacillus subtilis

The antibacterial test results for the secretion of the five Chinese fir cultivars against B. subtilis are shown in Fig. 2. The exudates of the cultivars exhibited significant antibacterial effects against B. subtilis (P < 0.05). The average diameter of the inhibition zone for Yangkou 3, Yang 020, Red-heart Chinese fir, Yang 061, and Chuizhisha was 2.70, 2.48, 3.07, 2.56, and 1.90 times that for CK, respectively. The bacterial cells displayed moderate sensitivity toward Taxodium distichum but high sensitivity toward the other four secretions. The antibacterial effect of the exudates of the five Chinese fir cultivars was in the following order: Red-heart Chinese fir > Yangkou 3 > Yang 061 > Yang 020 > Chuizhisha.

Figure 2 Comparison of the antibacterial effect of wound secretions of different Chinese fir cultivars on Bacillus subtilis.

The asterisks (**) in the middle of the figure indicate significant differences between the Chinese fir and CK. Different lowercase letters indicate that there are significant differences in the antibacterial effects of exudates of different Chinese fir cultivars.

The MIC test results for the different secretions against B. subtilis are shown in Table 3. The MIC of Red-heart Chinese fir, Yangkou 3 and Yang 061, Yang 020, and Chuizhisha was 0.5, 1, 2, and 4 mg/mL, respectively. The MICs of exudates of Chuizhisha, Yang 020, and Yang 061 on B. subtilis were greater than those of Yangkou 3 and Red-heart Chinese fir. The MIC of exudates of Yangkou 3rd generation on B. subtilis was greater than that of Red-heart Chinese fir, and the exudates of Red-heart Chinese fir exhibited the strongest inhibitory effect on B. subtilis.

Table 3 Minimum inhibitory concentration of wound secretion from different Chinese fir cultivars against Bacillus subtilis..

Chinese fir cultivars	Secretion concentration (mg mL−1 )	
	0.25	0.50	1	2	4	8	
Yangkou 3	++	+	–	–	–	–	
Yang 020	+++	++	+	–	–	–	
Red-heart	+	–	–	–	–	–	
Yang 061	++	+	–	–	–	–	
Chuizhisha	+++	++	+	+	–	–	
Notes.

+++ indicates that the colony grows vigorously at this secretion concentration

++ indicates that the colony grows more at this secretion concentration

+ indicates that the colony grows a small amount at this secretion concentration

– indicates that the colony grows aseptically at this secretion concentration

Antibacterial effect of wound secretions from different Chinese fir cultivars on Salmonella paratyphi B

The antibacterial test results for the secretions of different Chinese fir cultivars against S. paratyphi B are shown in Fig. 3. All exudates exhibited significant antibacterial effects against S. paratyphi B (P < 0.05). The average diameter of the inhibition zone for Yangkou 3, Yang 020, Red-heart Chinese fir, Yang 061, and Chuizhisha was 2.21, 1.72, 2.00, 2.00, and 1.83 times that for the CK, respectively. The exudates exhibited moderate antibacterialactivity on S. paratyphi B, and their antibacterial effects were in the following order: Yangkou 3 > Red-heart Chinese fir > Yang 061 > Chuizhisha > Yang 020.

Figure 3 Comparison of the antibacterial effect of wound secretions of different Chinese fir cultivars against Salmonella paratyphi B.

The asterisks (**) in the middle of the figure indicate significant differences between the Chinese fir and CK. Different lowercase letters indicate that there are significant differences in the antibacterial effects of exudates of different Chinese fir cultivars.

The MIC test results for the secretions of different Chinese fir cultivars against S. paratyphi B are shown in Table 4. The MICs of Yangkou 3, Red-heart Chinese fir, and Yang 061 were 2 mg/mL, and those of Yang 020 and Chuizhisha were 4 mg/mL. The MICs of the exudatesof Chinese fir Yang 020 and Chinese fir on S. paratyphi B were greater than those of Yangkou 3, Red-heart Chinese fir, and Yang 061, with Yangkou 3 exhibiting the strongest antibacterial effect on S. paratyphi B.

Table 4 Minimum inhibitory concentration of wound excretion from different Chinese fir cultivars against Salmonella paratyphi B.

Chinese fir cultivars	Secretion concentration (mg mL−1 )	
	0.25	0.50	1	2	4	8	
Yangkou 3	++	+	+	–	–	–	
Yang 020	+++	++	+	+	–	–	
Red-heart	++	+	+	–	–	–	
Yang 061	++	+	+	–	–	–	
Chuizhisha	+++	++	+	+	–	–	
Notes.

+++ indicates that the colony grows vigorously at this secretion concentration

++ indicates that the colony grows more at this secretion concentration

+ indicates that the colony grows a small amount at this secretion concentration

- indicates that the colony grows aseptically at this secretion concentration

Antibacterial effect of wound secretions from different Chinese fir cultivars on Pseudomonas aeruginosa

The antibacterial test results of the secretions of different Chinese fir cultivars against P. aeruginosa are shown in Fig. 4. All exudates exhibited significant antibacterial effects against P. aeruginosa (P < 0.05). The average diameter of the inhibition zone for Yangkou 3, Yang 020, Red-heart Chinese fir, Yang 061, and Chuizhisha was 1.70, 1.35, 1.93, 1.83, and 1.51 times that for the CK, respectively. The inhibition effect was moderately sensitive, with an average diameter of the inhibition zone of 9.07 mm, indicating a relatively lower level of sensitivity. The antibacterial effect of the secretions was in the following order: Red-heart Chinese fir >Yang 061 > Yangkou 3 >  Chuizhisha > Yang 020.

Figure 4 Comparison of the antibacterial effect of wound secretions of different Chinese fir cultivars against Pseudomonas aeruginosa..

The asterisks (**) in the middle of the figure indicate significant differences between the Chinese fir and CK. Different lowercase letters indicate that there are significant differences in the antibacterial effects of exudates of different Chinese fir cultivars.

The MIC test results for the secretions of the different cultivars of Chinese fir against P. aeruginosa are shown in Table 5. The MIC of Yangkou 3, Hongxinshan, and Yang 061 was 4 mg/mL, whereas that of Yang 020 and Chuizhisha was 8 mg/mL The MICs of the secretions of Yang 020 and Chuizhisha on P. aeruginosa were greater than those of Yangkou 3, Red-heart Chinese fir, and Yang 061.

Table 5 Minimum inhibitory concentration of wound excretion from different Chinese fir cultivars against Pseudomonas aeruginosa.

Chinese fir cultivar	Secretion concentration (mg mL−1 )	
	0.25	0.50	1	2	4	8	
Yangkou 3	+++	++	+	+	–	–	
Yang 020	+++	+++	++	+	+	–	
Red-heart	+++	++	+	+	–	–	
Yang 061	+++	++	++	+	–	–	
Chuizhisha	+++	+++	++	++	+	–	
Notes.

+++ indicates that the colony grows vigorously at this secretion concentration

++ indicates that the colony grows more at this secretion concentration

+ indicates that the colony grows a small amount at this secretion concentration

– indicates that the colony grows aseptically at this secretion concentration

Antibacterial effect of wound secretions from different Chinese fir cultivars on Escherichia coli

The antibacterial test results for the secretions of the different Chinese fir cultivars against E. coli are presented in Fig. 5. All exudates exhibited antibacterial effects against E. coli. The average diameter of the inhibition zone for Yangkou 3, Yang 020, Red-heart Chinese fir, Yang 061, and Chuizhisha was 2.68, 2.87, 2.79, 2.92, and 2.91 times that for the CK, respectively. The secretions of the five Chinese fir cultivars exhibited a high antibacterial effect on E. coli. The antibacterial effects of the secretions on E. coli were in the following order: Yang 061 > Chuizhisha > Yang 020 > Red-heart Chinese fir > Yangkou 3.

Figure 5 Comparison of the antibacterial effect of wound secretions of different Chinese fir cultivars against Escherichia coli.

The asterisks (**) in the middle of the figure indicate significant differences between the Chinese fir and CK. Different lowercase letters indicate that there are significant differences in the antibacterial effects of exudates of different Chinese fir cultivars.

The MIC test results for the different secretions of the Chinese fir cultivars against E. coli are shown in Table 6. The MICs of Yang 020, Red-heart Chinese fir, Yang 061, and Chuizhisha were 0.5 mg/mL, and that of Yangkou 3 was 1 mg/mL. The MIC of exudates of Yangkou 3 on Escherichia coli was greater than that of Yang 020, Red-heart Chinese fir, Yang 061, and Chuizhisha. This result indicates that Yangkou 3 had the weakest antibacterialeffect on E. coli.

Antibacterial effect of wound secretions from different cultivars of Chinese fir against Staphylococcus aureus

The antibacterial test results for the secretions of the different Chinese fir cultivars against S. aureus are shown in Fig. 6. All secretions exhibited antibacterial effects against S. aureus. The average diameter of the inhibition zone for Yangkou 3, Yang 020, Red-heart Chinese fir, Yang 061, and Chuizhisha was 2.49, 2.55, 2.41, 2.44, and 2.53 times that for the CK, respectively. Notably, S. aureus exhibited moderate sensitivity towards Yangkou 3, Red-heart Chinese fir, and Yang 061, but high sensitivity towards Yang 020 and Chuizhisha. The antibacterial effect of the secretions on S. aureus was in the following order: Yang 061 > Chuizhisha > Yang 020 > Red-heart Chinese fir > Yangkou 3.

Figure 6 Comparison of the inhibition zone of wound secretions of different Chinese fir cultivars against Staphylococcus aureus..

The asterisks (**) in the middle of the figure indicate significant differences between the Chinese fir and CK. Different lowercase letters indicate that there are significant differences in the antibacterial effects of exudates of different Chinese fir cultivars.

Table 7 shows the MIC test results for the secretions of the different Chinese fir cultivars against S. aureus. The MIC of Yang 0 and that of Yangkou 3, Red-heart Chinese fir, Yang 061, and Chuizhisha was 2 mg/mL. The MIC of Yang 020 exudates against S. aureus was lower than that of the secretions of the four other Chinese fir varieties.

Table 6 Minimum inhibitory concentration of wound secretion from different Chinese fir cultivars against Escherichia coli.

Chinese fir cultivar	Secretion concentration (mg mL−1 )	
	0.25	0.50	1	2	4	8	
Yangkou 3	++	+	–	–	–	–	
Yang 020	+	–	–	–	–	–	
Red-heart	+	–	–	–	–	–	
Yang 061	+	–	–	–	–	–	
Chuizhisha	+	–	–	–	–	–	
Notes.

+++ indicates that the colony grows vigorously at this secretion concentration

++ indicates that the colony grows more at this secretion concentration

+ indicates that the colony grows a small amount at this secretion concentration

– indicates that the colony grows aseptically at this secretion concentration

Discussion

The extraction of natural active antibacterial substances from plants has attracted significant attention from scholars, and most plant extracts have inhibitory effects on B. subtilis, P. aeruginosa, E. coli, S. aureus B, etc. (Abdi & Dego, 2019; Pan & Huang, 2019). In the present study, our results confirmed our initial hypothesis (1) that the exudates of five Chinese fir varieties exhibit antibacterial effects on B. subtilis, S. paratyphi B, P. aeruginosa, E. coli, and S. aureus. However, the antibacterial effects varied among the different Chinese fir cultivars, both on the same bacterial species and across different bacterial species. The antibacterial effect of Red-heart Chinese fir on B. subtilis and P. aeruginosa was higher than that of other cultivars. The antibacterial effect of Yangkou 3 on S. paratyphi B was higher than that of other cultivars, but the antibacterial effect on E. coli and S. aureus was lower than that of other strains. The antibacterial effect of Yang 061 on S. aureus was higher than that of other cultivars. The antibacterial effect of Yang 020 on P. aeruginosa was lower than that of other cultivars. The antibacterial effect of Chuizhisha on B. subtilis was lower than that of other cultivars.

Table 7 Minimum inhibitory concentration of wound secretion from different Chinese fir cultivars against Staphylococcus aureus.

Chinese fir cultivar	Secretion concentration (mg mL−1 )	
	0.25	0.50	1	2	4	8	
Yangkou 3	++	++	+	–	–	–	
Yang 020	++	+	–	–	–	–	
Red-heart	++	++	+	–	–	–	
Yang 061	++	+	+	–	–	–	
Chuizhisha	++	+	+	–	–	–	
Notes.

+++ indicates that the colony grows vigorously at this secretion concentration

++ indicates that the colony grows more at this secretion concentration

+ indicates that the colony grows a small amount at this secretion concentration

– indicates that the colony grows aseptically at this secretion concentration

Considering that secretions are typically mixtures, the antibacterial effect resulting from the combination of antibacterial substances may exponentially increase or decrease (Marquardt et al., 2020). Shi et al. (2020) reported a significant improvement in the antibacterial effect upon combining different concentrations of different antibacterial extracts, indicating that the compounds exhibited a synergistic antibacterial effect. Chen et al. (2021) revealed an inhibitory effect of essential oils on different bacterial strains using the minimum inhibitory test on 18 types of plant essential oils. In the present study, the MICs of the exudates of different Chinese fir varieties on the five test bacteria differed. The average MIC was the highest for B. subtilis and the lowest for P. koraiensis. The MICs on S. paratyphi B and P. aeruginosa were in the following order: Yang 020 and Huizhisha > Yangkou 3, Red-heart Chinese fir, and Yang 061. The MIC for Yangkou 3 was higher than that of the other four cultivars on E. coli. The MICs for S. aureus were in the following order: Yang 061, Chuizhisha, Yangkou 3, and Red-heart Chinese fir > Yang 020. Notably, among the testes bacteria, the antibacterial effect of the secretions was the weakest on P. aeruginosa and the strongest on E. coli. This result is attributed to the inherent drug resistance of P. aeruginosa, a commonly acquired pathogenic bacteria (Wang et al., 2020; Cosentino, Viale & Giannella, 2023). Moreover, P. aeruginosa possesses an adaptation mechanism and robust metabolic ability, shielding it from external damage. The potent inhibition on E. coli may be attributed to the presence of aromatic and sesquiterpene compounds in the secretions (Nabi et al., 2022; Ban et al., 2023; Hu et al., 2023), which also contribute to the strong antibacterial effect of the exudates of Chinese fir.

We have previously demonstrated a significantly higher number of exudates in Chuizhisha than that of other Chinese fir varieties, with a significantly distinct composition compared to that of the other three (Yang 020, Yangkou 3, and Red-heart Chinese fir) varieties (Zeng et al., 2023).Our results substantiate the initial hypothesis (2) that cultivars with similar composition of exudates may exhibit similar antibacterial abilities. Despite Chuizhisha having a significantly higher number of exudates compared to other C. lanceolata varieties, the antibacterial effect of its exudates on B. subtilis, S. paratyphi B, and P. aeruginosa was poor. In contrast, the Red-heart Chinese fir with the least number of exudates exhibited the best antibacterial effect on these three bacterial species, indicating that the antibacterial effect of exudates may not be directly related to their number. Both the composition and antibacterial effects of exudates of Chuizhisha and Yang 061 are considerably similar, indicating that the antibacterial ability may be determined by the type of secretion rather than the number of secretions. Chuizhisha and Yang 061 exhibited the most potent antibacterial effect on E. coli and S. aureus. The exudates of these two Chinese fir cultivars are mainly composed of shikimic acid, isovitexin, and isopimaric acid, most of which are organic acid compounds. Consistent with our results, a previous study has demonstrated that the antibacterial properties of different substances vary; an antibacterial mixture based on organic acids exhibits a strong antibacterial effect against Escherichia coli (Corcionivoschi et al., 2023). The exudates in the Yangkou 3, Yang 020, and Red-heart Chinese fir exhibited the highest proportion of carnosol, accounting for more than 55% of the total, predominantly terpenoids. These secretions exhibited the most potent antibacterial effect on B. subtilis, indicating that carnosol is an effective antibacterial substance against B. subtilis, aligning with previous results (Pavić et al., 2019). The antibacterial effect of Yang 020 and Chuizhisha on Staphylococcus Aureus was significantly higher than that of Red-heart Chinese fir, which may be related to the difference of isovaleric acid in wound secretion (Zeng et al., 2023).

This is the first antibacterial study on exudates of Chinese fir and it helps us to screen Chinese fir cultivars with high-disease-resistant. However, because of the nature of the experimental materials and insufficient concentrations used, the antibacterial effects of Chinese fir secretions on bacteria could not be elucidated comprehensively. Additional studies are required to compare the effects of the secretions of Chinese fir from forests with different ages and provenances, in addition to comparing the material contents and antibacterial effects of different fir wound secretions, to explore the antibacterial effects of Chinese fir secretions on bacteria comprehensively.

Conclusions

Our results confirmed our initial hypothesis (1) that the exudates of different cultivars of Chinese fir bark exhibited potent but varying antibacterial effects on B. subtilis, S. paratyphi B, P. aeruginosa, E. coli, and S. aureus. Moreover, the MIC for the different secretions on the five bacterial species differed; the average MIC was the highest for P. aeruginosa and the lowest for E. coli was the lowest. Our results also confirm our initial hypothesis (2) the antibacterial ability of the exudates is closely related to the composition of wound secretions, and the correlation with their quantity is unclear. Organic acid compounds may contribute to the antibacterial effect against E. coli, whereas terpenoids may contribute to the antibacterial effect against B. subtilis. Our findings provide important insights into the interaction between wound secretion in trees and different bacterial species, which can expand the use of Chinese fir wound secretions, and also offering a theoretical basis for screen Chinese fir cultivars with high-disease-resistant.

In summary, this study compared and discussed the antibacterialeffect of the exudates of five Chinese fir cultivars. However, due to the limited experimental materials (five cultivars), the mechanism underlying the antibacterialeffect of these secretions was not fully revealed. Follow-up studies are required to compare firs of different ages and provenances, further exploring the antibacterial effect of Chinese fir secretions (Jiang et al., 2023).

Supplemental Information

Supplemental Information 1 Antibacterial effect data of wound secretions of different strains of Chinese fir

Each data represents a measurement result.

Supplemental Information 2 The code of box line diagram

Supplemental Information 3 LSD

We are grateful for the experimental materials provided by Yangkou State-owned Forest Farm.

Additional Information and Declarations

Competing Interests

Author Contributions

Data Availability

The authors declare there are no competing interests.

Yu Jiang conceived and designed the experiments, performed the experiments, analyzed the data, prepared figures and/or tables, authored or reviewed drafts of the article, and approved the final draft.

Yalin Zeng conceived and designed the experiments, performed the experiments, analyzed the data, prepared figures and/or tables, and approved the final draft.

Jianing Zhu performed the experiments, prepared figures and/or tables, and approved the final draft.

Linjun Sun performed the experiments, prepared figures and/or tables, and approved the final draft.

Pengfei Wu conceived and designed the experiments, authored or reviewed drafts of the article, and approved the final draft.

Ming Li conceived and designed the experiments, authored or reviewed drafts of the article, and approved the final draft.

Xiangqing Ma conceived and designed the experiments, authored or reviewed drafts of the article, and approved the final draft.

The following information was supplied regarding data availability:

The raw measurements and the code are available in the Supplementary Files.

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
