# Peer review of "Comparative study of the antibacterial effects of wound secretions of different cultivars of Chinese fir"

_PeerJ, doi:10.7717/peerj.17850_

## Round 0.1 · original submission · Major Revisions

You are requesting to revise the manuscript addressing comments of the reviewers.

Reviewer 1 ·

Basic reporting

Comments
• Line 33: “The mean MIC…..” – Instead of saying “higher MIC” , you can say “lower potency”. That makes more sense.
• Line 42: This is incomplete statement. This study offers …..
• Line 51: et al must always be italicized, et al.
• Line 66: “However, there is a lack of….”. – This statement doesn’t seem correct.
It must be “However, there is a lack of……. different varieties and cultivars of Chinese firs.”
• Line 72- 78: “Bacillus subtilis……”. – The rationale of mentioning the characteristics of these bacterial species is not clear. The importance of B. subtilis in plant growth is mentioned, followed by infections caused by P. aeruginosa, E. coli and S. aureus. What is the motivation of the study? Do you want to determine if the Chinese fir secretions can be used to treat human bacterial infections or how these secretions make Chinese firs resistant to disease?
• Line 79: “In the 1960s, …..”- Original publication is appreciated.
• Line 94: “we hypothesized…..”. Bacteriostatic and antibacterial are not synonyms. Bacteriostatic is inhibition of growth.
• Line 108: “Yankou 3 is a seedling…..” Please mention the timeline, if you want to. Third generation does not mention how old the seedling was.
• Line 127: “The collected wound secretions….” – This statement can be modified as “The antibacterial effects of the collected wound secretions of Chinese firs were to test against different bacterial species.”
• Line 138: “The bacterial stock solution….” – Did the bacteria grow in the sterile water?
• Line 137- 147: The methodology is no clear. This needs revision. How much the bacterial stock in ‘sterile water’ incubated for?
• Line 150: “Sterile filter paper infiltrated….” This statement can be modified as “sterile water soaked…”
• Line 158: “A total of 100 L…..” This is not clear. Line 144 mentions bacterial culture is diluted to two different concentrations. Was this culture further diluted by adding 100 L to 9 mL sterile water? Please keep the methodology in continuation.
• Line 162: “Each concentration gradient….” Since you plate very high number of bacetria on the plate, I don’t expect isolated colonies. So, the ‘presence of growing colonies’ is a difficult measure. Instead, the statement can be modified as “Growth inhibition was used as to determine the antibacterial activities of the wound secretions.”
• Line 247: All the bacterial names must always be italicized. Once the bacterial species in mentioned in the article, after that all the bacterial names can be mentioned as P. aeruginosa, E. coli etc.
• Discussion must be different from results. This discussion paragraph is similar to results section. Line 280-305 fits into the discussion very well.
• Generally, Mueller-Hinton agar is used to test the antibacterial effects of the unknown compounds. Please clarify why MH medium was not used.

Experimental design

Methodology of the compounds testing was done is not clear. Attention needs to be given to the details like how many hours incubation was done, how was bacteria diluted and to what concentration etc.
Rationale of using the bacterial species is not convincing.

Validity of the findings

Triplicates have been performed. But validity needs more better understanding of the methods. Methods are not very concise and clear

Reviewer 2 ·

Basic reporting

The manuscript is fairly easy to understand but its organisation of the results section can be improved. The manner by which the results are presented in Tables 3-7 can be improved. There is no need to use a separate table for each bacterial species.

Please consider to include the following:
1. An overall experimental design (with details on the number of trees, secretions collected, and how these are used in the assay)
2. A representative photograph of how the 'wound secretions' were collected from the trees.

Experimental design

Standard methodologies were used but details are missing in most parts.

Important: The proper scientific term should be used to refer to the "wound secretions". I am not sure if "exudates" should be used.

1. There are 5 cultivars and 15 trees were selected (line119). Please clarify the overall experimental design. Are the 15 trees consists of all 5 cultivars? Did the authors separate the exudates from each tree or combine them? If it is the latter, how it was done?

2. Please state the storage conditions of the exudates between collection and use in the antibacterial assays. How long were the exudates kept before they are used in the assays?

3. The concentration of exudates from different trees, different cultivars or even collection on different days is likely to be different but the data in Tables 3-7 seemed to suggest that the authors were able to dilute them to various concentrations. Please explain.

4. Minimum inhibitory concentration (MIC) is defined as the lowest concentration of an antimicrobial agent that will inhibit the visible growth of a microorganism after overnight incubation. But data in Tables 3-7 are not single concentrations. Please clarify.

5. An appropriate positive control could have been included to allow comparison to be made.

This manuscript fails to meet PeerJ's standard in this area.

Validity of the findings

The research objective of this study is interesting but unclear experimental design and the lack of details limit our interpretation of the findings. Some of the issues to be addressed are as follows:

It was stated that "the antibacterial ability of the wound secretions is closely related to the composition of wound secretions" (lines 320) but there is no data presented to the chemical composition of the exudates. Not even simple phytochemistry testing or TLC to give us insights on the nature of the major compounds. Hence, it is difficult to convince the readers that there is indeed a (significant) difference in the chemical profiles.

The antibacterial effect of the exudates has to be interpreted with care, comparison with an appropriate positive control or material of similar nature will be best. Then the results can be used as a basis for further research into the mechanistic action or factors that can affect the production.

This manuscript fails to meet PeerJ's standard in this area.

Additional comments

This study explores the potential use of the 'wound secretions' or exudates of the Chinese fir bark as potential antibacterial agent which no doubt is interesting based on the possible ecological roles of these secretion but its feasibility for the commercial application need to be considered. The findings presented in the manuscript is rather preliminary. The experimental design is weak with many details not provided. The choice of the bacterial species also not given. The secretions were found to demonstrate varying % of inhibitory action on the bacterial species but there is no reference to any positive controls, so its efficacy cannot be deduced. The chemical profiles of the secretions should be carried out and presented, so the relationship (if any( between the different secretions and their antibacterial activity can be established.

---

## Round 0.2 · accepted · Accept

Accepted and now ready to go for editing.